# A New Method to Monitor the Nutritional Quality of Packaged Foods in the Global Food Supply in Order to Provide Feasible Targets for Reformulation

**DOI:** 10.3390/nu13020576

**Published:** 2021-02-09

**Authors:** Fabienne Leroy, Andreas Rytz, Adam Drewnowski, Marie Tassy, Audrey Orengo, Veronique Rheiner Charles, Hilary Green

**Affiliations:** 1Nestlé Research, Vers-chez-les-Blanc, 1000 Lausanne 26, Switzerland; fabienne.randall@gmail.com (F.L.); andreas.rytz@rdls.nestle.com (A.R.); audrey.orengo@rdls.nestle.com (A.O.); veronique.rheiner-charles@rdls.nestle.com (V.R.C.); j.hilary.green@gmail.com (H.G.); 2Center for Public Health Nutrition, Department of Epidemiology, School of Public Health, University of Washington, Seattle, WA 98195-3410, USA; adamdrew@uw.edu

**Keywords:** public health, nutrients, food supply, packaged food

## Abstract

Nutrient profiling systems, initially designed to promote healthy food choices at the point of sale, can also provide the scientific basis for innovation and product reformulation by the food industry. This work presents a new profiling system to help define feasible nutrient targets for reformulation of packaged foods. The focus is on five key nutrients for which the World Health Organisation (WHO) has set population-level goals: sugar, saturated fat, sodium, fiber, and protein. The methodology uses Mintel’s Global New Products Database of packaged foods to (1) identify nutrients relevant to each food category (2) sort products into sub-categories defined by a unique nutritional signature, and (3) develop standards for “best of class” products. For instance, if targeted to be amongst the best 15% of the global food supply, pizza must have less than 4.0 g/100 g saturated fat, less than 520 mg/100 g total sodium and more than 9.8 g/100 g protein. Fiber and sugar are not identified as relevant nutrients for the pizza category and no targets are provided.

## 1. Introduction

Nutrient profiling has been defined by the World Health Organization (WHO) as “the science of classifying or ranking foods according to their nutritional composition for reasons related to preventing disease and promoting health” [1]. Nutrient profiling of foods was initially intended to promote compliance with dietary guidelines by steering consumers toward healthier food choices. Models developed in the UK [2], France [3], the US [4], and the Netherlands [5] became the basis for more than a hundred nutrient profiling models that are now in existence worldwide [6]. Developed by governments [7], international agencies [8,9] and the private sector [10], such models serve a variety of public health goals, providing the scientific basis for front-of-pack labeling [7], taxation initiatives [6], and regulating marketing and advertising to children [11]. Nutrient profiling, when used to support the implementation of dietary guidelines can be an effective educational and policy tool [6].

Current nutrient profiling models are distinct from single nutrient content claims (“a good source of protein”) or nutrient related warning labels (“high in fat”; “high in sugar”) [12]. Such product health claims are typically under the direct jurisdiction of national or regional regulatory agencies such as the Food and Drug Administration in the US or the European Food Safety Authority in the European Union (EU) [2]. Rather, the goal of voluntary nutrient profiling models is to capture the overall nutrient density of a given food using a complex formula and then convey that information to the interested consumer. To do this, nutrient profiling schemes have expressed the overall nutrients density of a foods as a letter grade [7], a number [13], or a logo [14,15,16].

Most early nutrient profiling schemes were aimed at supermarkets and the individual consumer. None were specifically developed to provide food producers with nutrient targets for product reformulation. Previous studies identified very few models (6 out 78) as having product reformulation as a possible application, but none had reformulation as a primary objective [6]. Among those, none were entirely suitable for setting nutrient targets for the reformulation of processed foods. To the contrary, some of the standards set by nutrient profiling models appeared to be more aspirational than realistic. Some standards were narrowly applied to selected product categories (“non-essential foods”), selected nutrients (fat, sugar, and salt) and selected age groups. The use of nutrient profiling models to shape food demand could be extended to the shaping of the global food supply.

The aim of the present paper is to provide a new way of assessing the overall nutritional quality of packaged foods in the global food supply, in order to provide feasible nutritional targets for reformulation. This method will also allow food companies to benchmark the nutritional quality of their products against that of competitors. Better ways to assess nutrient density of company products portfolios will allow governments and public health authorities to monitor changes in nutritional quality of population diets over time. Improving the healthfulness of the food supply through product reformulation is one goal of nutrition-related policies and programs. With the help of new nutrient profiling models, product reformulation could become a key strategy for health promotion at the population level.

## 2. Materials and Methods

### 2.1. New Nutrient Profiling Model

The development of the new nutrient profiling model followed previously published steps [17].

#### 2.1.1. Components

The model uses five nutrients that have been identified as relevant for public health [18]. These are:Three disqualifying components (saturated fat, total sugars, total sodium)Two qualifying components (protein and dietary fiber)

#### 2.1.2. Reference Unit

The model evaluates product composition as grams per 100 g/100 mL and normally as sold. In the case of powder products that must be reconstituted prior to consumption, a constant dilution factor of 1 g powder for 10 mL water was applied in order to assess these products as liquids and not as solids, since no product-specific dilution factor was provided in the food supply database.

Whenever available in Codex, “low in” and “source of” limits were chosen for upper and lower nutrient cut-points [19]. This applies for saturated fats, total sodium, and protein (with 50 g as the nutrient reference value for protein). For sugars and fibers, European regulation was applied [20]. Fiber limit for liquids was set as being half of the value for solids (Table 1).

#### 2.1.3. Model Outline

The model generates nutrient thresholds in the reference unit, which are single values for each of the nutrients in the model. They serve as targets for reformulation, meaning that packaged foods must not exceed (upper limit), or must reach a given nutrient value (lower limit). Thresholds are derived from a food supply database featuring nutrient composition of products; the new algorithm considers nutrients simultaneously, not independently.

The model provides thresholds that are category specific. Relevant categories are derived from a food supply database using a new categorization algorithm.

The model applies to all packaged food and beverage products that can be reformulated, meaning that the content of one or more of the five considered nutrients can be improved during manufacturing. Consequently, products such as plain water, plain coffee, frozen fruits and vegetables without additions, or 100% juices are not rated by the model. Additionally excluded from model ratings were highly regulated products, such as infant formula and baby foods, as well as products developed for specific medical or technical purposes, such as food supplements or additives, enteral formulas, thickeners, or binders.

### 2.2. Food Supply Database

The Mintel Global New Products Database compiles nutrient content (including the five nutrients of interest) of all new launches of branded packaged foods and beverages in 62 countries worldwide [21]. These individual, exhaustive and dynamic data allow us to define any desired scope [22]. Using entries from three consecutive years ensures up to date, qualitative and representative data that are categorized according to the consumer offering. This categorization does not follow strict nutritional criteria. As an example, the Mintel “dairy” category covers sub-categories such as “white milk” and “flavored milk”, but it also covers “plant based drinks” that are not dairy but are presented as plant based milk “alternatives”. Mintel data also place butter and margarine in the dairy category. Dairy products that do not contain calcium are not part of the dairy group in the US and are classified as fats. This type of categorization enables the comparison of products within a same consumer offering.

Once cleaned, the database for the period 2016–2018 features 19 Mintel categories, 107 eligible Mintel food sub-categories, and 442,018 individual products. Cleaning consisted of removing products for which the sum of nutrient content values per 100 g exceeded 100 g. In cases where salt content was available, but sodium was not, sodium content was estimated by multiplying the salt content by the ratio of the molecular weights of sodium (22.99 g/mol) and sodium chloride (55.44 g/mol).

Data for all five nutrients of interest were available for 174,382 products (39%) only. Protein was declared most often (97%), followed by total sodium (89%), saturated fat (78%), total sugars (77%), and fiber (49%). Missing data were strongly related to regional policies, for example Russia and Ukraine declare protein only, and to product sub-categories, with fiber being declared for 92% of “cold cereals” and 16% of “carbonated soft drinks”. The highest number of protein omissions was also observed in the category "carbonated soft drinks" (18%) which was expected since carbonated soft drinks usually do not contain fibers or proteins. Therefore, missing values are informative and considered as such in all product categories (i.e., neither set to zero, nor imputed to any value).

Among the 427,357 products featuring data for protein (97%), values spread between 0 and 100 g/100 g, with 10% values being equal to 0 g/100 g, 50% lower than 7 g/100 g and 98% lower than 30 g/100 g (Figure 1A). This large spread is strongly related to product categories as illustrated for “margarine and other blends” (0–4 g/100 g), “pizza” (5–15 g/100 g), and “hard cheese and semi hard cheese” (>16 g/100 g) (Figure 1B).

Since product category heavily impacts both missing nutrient content and nutrient distributions, the new nutrient profiling model aims at identifying nutrient thresholds that are category specific. The considered product categories are the 107 sub-categories present in Mintel. For some of these categories, the declared unit heavily depends on regional habits and policies. This is the case for ice-creams for which all nutrients are predominantly declared as g/100 g in America but g/100 mL in Asia and Europe. It is also the case for reconstituted powders (e.g., dry soups and beverages) for which all nutrients are predominantly declared as g/100 g of powder in Asia and America and as g/100 mL of reconstituted beverage in Europe. To account for these differences, such sub-categories are split, which leads to 118 initial categories featuring the 442,018 considered products (Appendix A).

### 2.3. Algorithms to Define Relevant Nutrient Thresholds by Product Category

The proposed model can be used to rank products within their category according to overall healthiness, as defined by their nutritional composition. This ranking can be used to define the nutrient composition required for a product to be in the healthiest x percent of all products on the market. To do this, three algorithms were developed to define nutrient thresholds by product category.

Identification of category-relevant nutrientsRefinement of the initial product categorizationDerivation of category-specific nutrient thresholds that can be used as targets for product formulation.

These algorithms are illustrated using the 968 products in the category of “margarine and other blends”, a category for which only two nutrients are identified as relevant and from which only three nutritionally homogeneous sub-categories were derived. This example is chosen to provide a simple visualization in two dimensions of the method (Figure 2). However, the algorithms work the same way for categories with more than two relevant nutrients or with a different number of nutritionally homogeneous sub-categories.

#### 2.3.1. Identification of Category-Relevant Nutrients

The selection of category-relevant nutrients is based on nutrient distributions. A nutrient is considered category-relevant if more than half of the products within this category have a declared nutrient content with a value that exceeds the “low in” or the “source of” limit (Table 1). In the example of “margarine and other blends”, this is the case for saturated fat and total sodium only (Figure 2A).

With this selection, it is possible to keep 850 out of 968 products (88%) that feature values for both saturated fat and total sodium, regardless of the missing values of other nutrients. Using the same algorithm, category-relevant nutrients are identified for 112 out of 118 initial categories and 350,994 products (79%) are kept for further analyses. The six initial categories for which no nutrient was identified as category-relevant feature 8732 products (<2%) and cover product categories such as water or medicated confectionery that were borderline in-scope (Appendix A).

The initial category “margarine and other blends” is the only one, among the 19 belonging to the overall Mintel category “Dairy”, for which both saturated fat and total sodium are selected but none of the three other nutrients. This illustrates the relevance of the Mintel categorization that splits food group categories into nutritionally more homogeneous sub-categories.

#### 2.3.2. Refinement of the Initial Product Categorization

Some of the 112 categories are still nutritionally heterogeneous. This is problem because such categories group multiple consumer offering. As an example, the category “carbonated soft drinks”, for which only total sugars is a category-relevant nutrient, features products that are sweetened with either sugars or intense sweeteners. These two offers should be split to properly guide product reformulation. An algorithm was developed to automatize this split for all 112 initial categories, whenever relevant. This algorithm analyzes the univariate distributions of the selected category-relevant nutrients. The category is considered homogeneous at nutrient level if the distribution was unimodal; it is considered heterogenous and potentially composed of multiple offers if the distribution was multimodal, with separate offers for each mode. In this case, the initial category is split.

As an example, the bivariate and univariate distributions of saturated fat and total sodium are represented for the initial category “margarine and other blends” (Figure 2B). Although the distribution of saturated fat covers a large range, from less than 5 g/100 g to more than 50 g/100 g, it is unimodal with a peak around 12 g/100 g. This indicates that there is no need to split the category with respect to saturated fat. The distribution of total sodium also covers a very large range, from 0 to more than 1000 mg/g. It is furthermore trimodal and the category should be split into three offers, namely “low salt” grouping 267 products with less than 270 mg/100 g, “medium salted” products grouping 293 products with 270–540 mg/100 g and “highly salted” products grouping 290 products with more than 540 mg/100 g total sodium.

In this example, only one of the two category-relevant nutrients presented multimodality and the number of final categories was therefore three (1 × 3 = 3). If saturated fat were also multimodal, say bimodal, the number of final categories would simply be six (2 × 3 = 6).

The aim is not to artificially multiply the number of categories, but to account for different product offers. The multimodality detection algorithm must therefore only identify major modes and be less sensitive than traditional tests for modality [23]. The proposed algorithm considers a histogram with exactly 12 bins, each of them being considered a potential mode if it reveals convexity, meaning that it counts more products than the average of its two neighbor bins. If a bin is considered a potential mode, but none of its neighbors is, the mode is set as the value in the middle of that bin. If neighbor bins were considered potential modes, the mode is set as the value in the middle of all these neighboring bins. Finally, the category is split using the value in the middle of two consecutive modes as cutoff to differentiate products. As an example, among the 12 bins of the histogram of total sodium content in “margarine and other blends” (Figure 2B), he bins 90–180, 360–450, and 630–720 qualify to feature a mode, but none of their neighbor bins. Consequently, modes are set respectively at 135, 405, and 675 mg/100 g and the category is split at respectively 270 (mean of 135 and 405), and 540 (mean of 405 and 675) mg/100 g

With this method, the 112 initial categories with at least one category-relevant nutrient are split into 263 categories, with 59 initial categories staying unchanged, 48 initial categories being split into two, three or four final categories and 5 categories that are split into more final categories. These latter very heterogeneous categories cover “cold cereals”, “nuts”, “popcorns”, or “snack cereal energy bars” that all feature products with a very high variety of possible inclusions and coatings that naturally lead to very heterogeneous nutrient profiles (Appendix A).

#### 2.3.3. Derivation of Category Specific Nutrient Thresholds

Thresholds are derived for each category-relevant nutrient by simultaneously considering all of them. The minimal common percentile approach is used [10]. As an example, the bivariate distribution of saturated fat and total sodium is represented for the category “margarine and other blends—highly salted” (Figure 2C). Thresholds of these two category-relevant nutrients are derived from this bivariate distribution. As an example, the 50% nutrient threshold corresponds, in this case, to the 65% percentile (P65%) on each of these nutrients. This means that 50% of the products are simultaneously lower than both limits, namely, 28.6 g/100 g saturated fat and 714 mg/100 g total sodium. These 50% products belong therefore to the better half of products with respect to all category-relevant nutrients. Consequently, these nutrient thresholds can guide formulation towards products belonging to the better half of the food supply. If more ambitious, one can derive nutrient thresholds to guide formulation towards the top 15% or towards any other target. The 15% nutrient thresholds are 17.9 g/100 g saturated fat and 643 mg/100 g total sodium.

In cases where the threshold of a category-relevant nutrient was below the limit “low in” or “source of”, the threshold is set to this limit and the algorithm is rerun so as to capture the expected proportion of products when considering all category-relevant nutrients.

### 2.4. Method to Assess the Validity of the New Nutrient Profiling Model

Different methods exist to test the validity and accuracy of a new nutrient profiling method (e.g., criterion, convergent, construct, and content) [24]. The convergent method aims at comparing the new Nutrient Profiling model with a previously validated profiling model.

As there is no consensus about the superiority of one particular nutrient profiling model, previous studies have screened 78 models and identified the Australian Health Star Rating (HSR) as a relevant reference [6].

The HSR has been developed in consultation with relevant stakeholders, it includes most food and beverage categories present in the general market and generates a classification into “healthy” and “unhealthy” foods that has been validated and that is consistent with general nutrition principles. The HSR furthermore differentiates nutritional quality within and between categories, is publicly available and gives an holistic view of the nutritional quality of the product [25]. The HSR is a front-of-pack labelling system designed to enable an easy and standardized comparison of packaged foods. It is based on a scoring algorithm that deducts points for disqualifying nutrients (overall energy, sodium, total sugar, and saturated fat) and adds points for qualifying nutrients and ingredients (protein, fiber, calcium if relevant, and fruit and vegetables). Products are classified in six categories and their nutritional composition assessed per 100 g or 100 mL. Scores obtained are converted to a “Health Star Rating” between ½ to 5 stars. The Australian government led the development of the methodology in consultation with industry, public health, and consumer groups. Products with a HSR ≥ 3.5 stars can be promoted as healthy [26].

Since the HSR delivers product ratings, and not nutrient thresholds, it was necessary to adapt the new methodology to also deliver ratings. This was done to facilitate comparisons between the two methods. Each product is given a score between 0 and 100% corresponding to the proportion of products in the category that are less nutritious than the actual product, based on the minimal common percentile approach. As an example, a product yielding a score of 87% is a product that complies with the 13% nutrient thresholds, but not with the 12% nutrient thresholds. HSR (0.5 to 5 stars) and the new score (0 to 100%) are derived for all considered products, but fruit and vegetable content could not be considered because they were missing in the Mintel Global New Product Database. Boxplots are used to visualize the distribution of the new continuous scores as function of the ordinal HSR and weighted Spearman correlation is used to qualify the relationship.

## 3. Results

Nutrient thresholds are available for all category-relevant nutrients of all 263 final categories. Among these categories, total sodium is identified most often as category-relevant (70%), followed by proteins (65%), saturated fat (51%), total sugars (47%), and fiber (32%). 

Reformulation of pizzas having been identified as a potential way of improving nutritional intakes in sensitive populations [27], nutrient thresholds are presented for the “pizza” category. In order to assess internal coherence of the method, these thresholds are further compared to those obtained for two categories that feature the main nutritional components (i.e., dough and cheese-topping) of a standard pizza, namely, “bread & bread products” and “hard cheese and semi-hard cheese”.

### 3.1. Results for Selected Product Categories: Bread, Cheese and Pizza

To be part of the better half of the food supply, bread products must have less than 560 mg/100 g total sodium and more than 7.6 g/100 g protein, cheese must have less than 21.4 g/100 g saturated fat, less than 788 mg/100 g total sodium, and more than 23.0 g/100 g protein, and pizza must have less than 5.0 g/100 g saturated fat, less than 600 mg/100 g total sodium, and more than 8.7 g/100 g protein. The 15% nutrient thresholds are given in addition to these 50% nutrient thresholds, together with the relative change that is needed to move from best 50% to best 15% of the food supply (Table 2).

These results show that the effort to move from an average product that meets the 50% nutrient thresholds to a top product that meets the 15% nutrient thresholds is not the same for all categories. Within a category, it might also be more challenging for a given nutrient, such as sodium in the case of cheese. Pizza yields nutrient thresholds that are fully in-line with those of its two main constituents, namely bread and cheese. Targeting sodium contents of 520 to 600 mg/100 g for pizza is therefore both feasible and relevant considering the intrinsic nature of the product. Targeting lower nutrient thresholds would be somewhat artificial because it would not be consistent with the category identity.

### 3.2. Relationship between HSR and the New Nutrient Profiling Model

The relationship between HSR (0.5 to 5 stars) and the new score (0 to 100%) is illustrated for the 3220 products of the pizza category (Figure 3). 

This figure shows strong agreement between the two methods, yielding a weighted Spearmann correlation r_WS_ = 0.81. The weighting accounts for the non-uniform HSR distribution with 4% pizzas yielding HSR ≤ 1.5, 19% HSR = 2, 6% HSR = 2.5, 48% HSR = 3, 22% HSR = 3.5, and 2% HSR > 3.5. 

The boxplots reveal that the vast majority of products rated HSR ≤ 1.5 yield a score lower than 10%, whereas a vast majority of products rated HSR ≥ 3.5 yield a score higher than 75%. On both ends, some individual discrepancies are nevertheless identified, with 3 products rated HSR ≤ 1.5 but yielding scores higher than 85% and 3 products rated HSR = 4 but yielding scores below 15%. The three products rated HSR ≤ 1.5 and scored higher than 85% are acceptable in terms of saturated fats, total sodium, and proteins but are extremely high in energy, which is related to high total fat content and which makes that HSR is also not positively accounting for the high protein content. On the opposite end, the three products rated HSR = 4 and scored lower than 15% are entirely vegetable based (dough included) and therefore do not provide enough protein to gain a high score.

This example is an illustration of the strong agreement between HSR and the new score. Consequently, the nutrient thresholds defined by new nutrient profiling model can serve as reformulation targets to further improve HSR score.

## 4. Discussion

The declared content of five key nutrients of 442,018 products launched between 2016 and 2018 is used to derive category-relevant nutrient thresholds of 263 homogeneous product categories, each corresponding to an existing nutritional offer in the food supply (Appendix A). The new method uses publicly available data and makes minimal assumptions, in order to set objective and feasible targets for product formulation. The proposed method offers the flexibility to manufacturing companies and policy makers to choose how high targets should be set relative to their strategy. The choice of the Mintel database, and some other methodological choices, nevertheless have an impact on both categorization and thresholds.

Although the new method could be applied to any database, results heavily depend on the product categorization of the actual database. Mintel’s categorization is optimal when aiming at guiding product (re)formulation because this database groups products according to global consumer offerings and therefore directly guides (re)formulation towards feasible, yet ambitious thresholds that are relevant within the competitive offering. In cases such offerings were not nutritionally homogeneous enough, the method refines the categorization according to nutritional criteria. As an example, carbonated soft drinks needed to be split into regular and diet alternatives to provide relevant thresholds for reformulation. More generally, the new method refines the categorization by analyzing the univariate nutrient distributions. This method eventually delivers less homogeneous clusters than multivariate clustering methods [28] but it is preferred for its ability to separate categories through independent cutoffs on all category-relevant nutrients.

Mintel covers only new launches; although, this is not necessarily representative of all products on-shelf, this certainly helps to set ambitious thresholds, because most companies strive for continuous improvement of product nutritional quality. 

Mintel’s coverage is limited to 62 countries. Moreover, due to regional labeling policies, there are inequalities in the availability of nutrient data. For example, more than 90% products are covered in Oceania, North America, and Western Europe, as well as for some Asian and South American countries. By contrast, less than 30% products are covered in China, Japan, Russia, and Ukraine, mainly due to policies limiting the number of nutrients to be declared. Although this is not ideal, the impact is limited since the total of these countries account for less than 10% for most product categories. In the few cases where these countries are dominant, such as for “stock others” (78%), “meat snacks” (47%), and “instant noodles” (39%), the nutrient selection is driven by them and becomes therefore again representative of available data.

More fundamentally, whatever the chosen database, the new method relies on data of products on the shelf; it is therefore more suited for renovation than for innovation. Moreover, when specific reconstitution guidelines are not provided for powdered products in the database, a constant dilution factor is applied to assess the intrinsic nutritional quality of the product. However, this might not fully reflect the actual nutrient content of powdered products when consumed.

Although the method is not bound to any number of key nutrients to consider, it relies on objective limits such as “low in” for disqualifying nutrients and “source of” for qualifying nutrients; these limits need therefore to exist for all considered nutrients. Furthermore, there is a tradeoff between the considered complexity of the nutritional profile and the availability of data. In this sense, the selection of the five key nutrients saturated fat, total sugars, total sodium, protein, and fiber is optimal. Furthermore, the logic to consider a nutrient as category-relevant if more than 50% products have a value higher than the nutrient content claim limit is also part of this tradeoff: this value of 50% simultaneously maximizes the number of category-relevant nutrients and the number of products to be kept for deriving thresholds. 

However, as shown for the Pizzas category, even if the number of category-relevant nutrients is lower than five, the ranking of products with the developed methodology is strongly correlated with the HSR rating of the same products. Apart from few exceptions, the new method gives similar results to a validated method such as HSR and therefore enables developers of manufactured food to target reachable limits for the nutrients that matter most in a category while ensuring the desired healthiness is achieved. Moreover, it shows that, because products within a category are often manufactured using comparable ingredients, it is enough to consider nutrient quantity, and not nutrient quality to assess the holistic nutritional quality of a food compared to its competitors. This high correlation of the new method with HSR provides first evidence that it is valid relative to an accepted methodology [29]. Further studies could aim at correlating the new method with other nutrient profiling models. 

The product categorization method is novel as it differs from previous profiling system by its objectivity. Most category-specific systems are based on a non-mathematical method of categorization, and therefore may be biased to obtain the desired results. On the contrary, the present method is based on the convexity of the nutrient distributions, and the only variable element is its sensitivity. Therefore, by varying the sensitivity of the categorization algorithm, this method could be adapted to other needs. A lower sensitivity could lead to fewer categories and allow the development of a system to guide the consumer to healthier products rather than guiding reformulation. Furthermore, if applied to a wider database including both commodities and packaged food, it could help create more homogeneous categories in order to develop population-based recommendations.

To verify the public health impact of newly developed reformulation targets, further research should model how dietary intakes would be modified when currently consumed products would be substituted by products reaching 50% thresholds in their category [27].

The present method could be adapted to tracking the nutritional value of newly developed food products as they enter the global marketplace. One challenge is how to address new product categories and potentially disruptive entries. For example, vegetables pizzas that score high on HSR do not contain much protein, and therefore have a very different nutritional signature form the traditional cheese pizzas. Consequently, they get low rankings in the current scheme. Identifying new product categories is a key component of ongoing innovation; at present some new products have marginal representation within Mintel database. Developing new categories for evaluation offers an opportunity to monitor the nutritional value of the global food supply.

## 5. Conclusions

The newly developed method is one of the first nutrient profiling system entirely data-driven able to provide objective and realistic nutritional targets for reformulation. It is useful to assess products’ nutritional quality against global competition and help set up nutritional criteria relevant at a global level for each food category representing a nutritionally homogeneous consumer offering. The method, applied on the Mintel database, shows strong agreement with Health Star Rating, and therefore, when implemented, will enable consumers to see that improvements have been made. Confirmatory analyses should aim at correlating the newly developed method with other nutrient profiling systems as well as modelling the potential impact of its implementation on nutrient intake at a population level. Further studies are also needed to make the approach sustainable in time and adaptable to food supply evolution.

## Figures and Tables

**Figure 1 nutrients-13-00576-f001:**
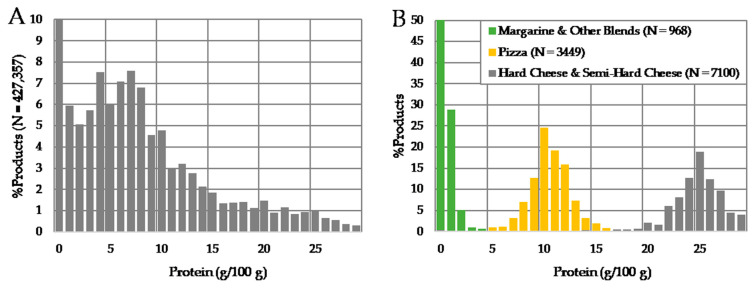
Relative distribution of protein among all N = 427,357 products featuring a protein content (**A**) and among three selected categories: Margarine and Other Blends (N = 968), Pizza (N = 3449) and Hard Cheese and Semi-Hard Cheese (N = 7100) (**B**).

**Figure 2 nutrients-13-00576-f002:**
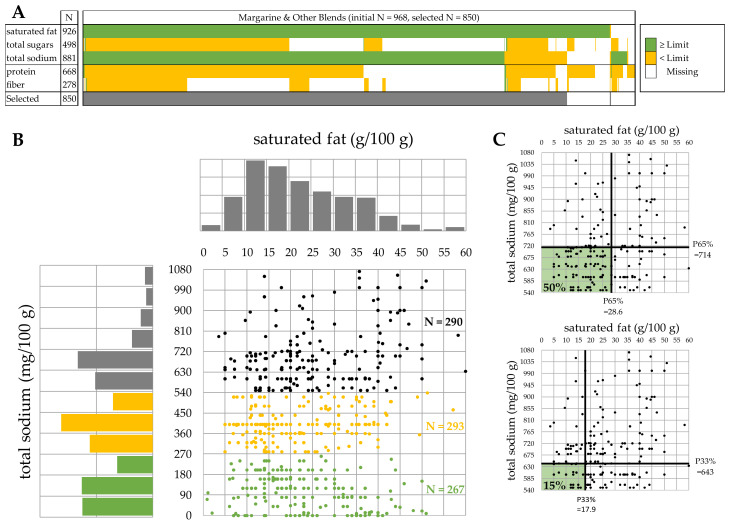
Illustration of the three algorithms using category “margarine and other blends”: identification of category-relevant nutrients (**A**), refinement of initial product categorization (**B**) and calculation of category specific nutrient thresholds for the “highly salted” category (**C**).

**Figure 3 nutrients-13-00576-f003:**
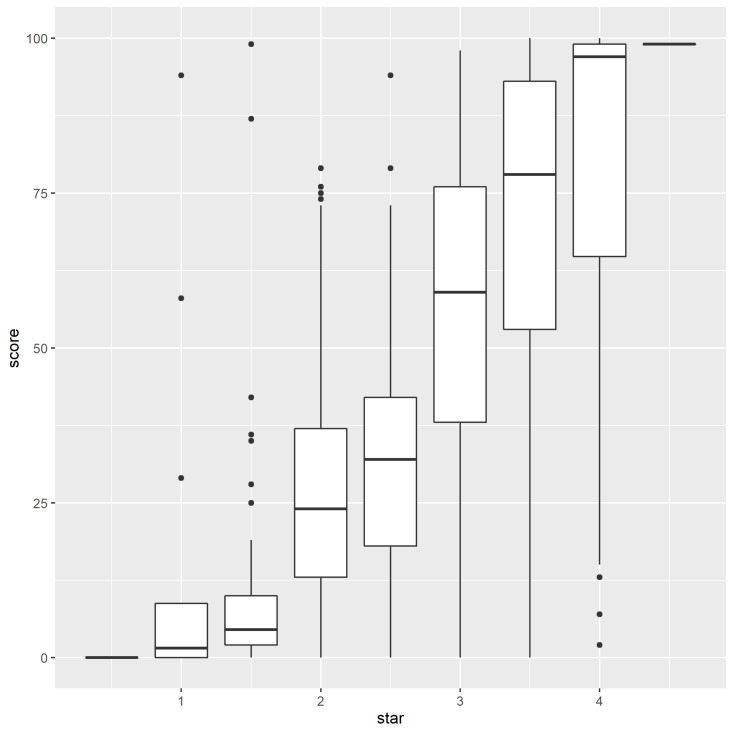
Boxplots visualizing the distribution of the new score as a function of the Health Star Rating (HSR) distribution in case of pizza category (N = 3220).

**Table 1 nutrients-13-00576-t001:** “Low in” limits for disqualifying components and “Source of” limits for qualifying components.

	Low in (if ≤ Limit)			Source of (if ≥ Limit)	
	Saturated Fat	Total Sugars	Total Sodium	Protein	Fiber
Solid (g/100 g)	1.50	5.0	0.120	5.0	3.0
Liquid (g/100 mL)	0.75	2.5	0.120	2.5	1.5

**Table 2 nutrients-13-00576-t002:** The 50% and 15% nutrient thresholds for “Bread and Bread Products”, “Hard and Semi-Hard Cheese”, and “Pizza” (with relative difference between the two thresholds in brackets).

		Saturated Fat (g/100 g)	Total Sodium (mg/100 g)	Protein (g/100 g)
Bread and Bread Products	50% Threshold	-	560	7.6
(N = 6532)	15% Threshold	-	412 (−26%)	9.3 (+22%)
Hard and Semi-Hard Cheese	50% Threshold	21.4	788	23.0
(N = 10,106)	15% Threshold	19.0 (−11%)	640 (−19%)	25.3 (+10%)
Pizzas	50% Threshold	5.0	600	8.7
(N = 3220)	15% Threshold	4.0 (−20%)	520 (−13%)	9.8 (+13%)

## Data Availability

Restrictions apply to the availability of these data. Data was obtained from 2019 Mintel Group Ltd and are available at https://www.mintel.com/global-new-products-database (accessed on 31 January 2019) with the permission of 2019 Mintel Group Ltd.

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
