# Peer review of "A New Method to Monitor the Nutritional Quality of Packaged Foods in the Global Food Supply in Order to Provide Feasible Targets for Reformulation"

_nutrients, 2021, doi:10.3390/nu13020576_

Round 1

Reviewer 1 Report

Manuscript (Nutrients-1062076) titled “A new method to monitor the nutritional quality of packaged foods in the global food supply in order to provide feasible targets for reformulation” aims to provide a new profiling system to define feasible nutrient targets for reformulation of packaged foods.

The manuscript is interesting; introduction is easy to read, methods are rather good, results and the discussions are clear.

In my opinion minor modifications are needed.

Lines 75-76: Why the authors use a dilution factor of 1g powder for 10ml water? There are powders that need to be diluted in 100ml water……  What do you means standard dilution factor?

Line 87: what does the nutrient threshold means? May be daily intake %? Explain better… 

Line 326: higher rather than higer  

Figure 2A: What do you means “not available” (white bar) 

Appendix A: there is a technical mistake:  you have capital letters into the table and roman numbers in caption

Author Response

Lines 75-76: Why the authors use a dilution factor of 1g powder for 10ml water? There are powders that need to be diluted in 100ml water……  What do you means standard dilution factor?

Thank you for raising this interesting point. Since no product-specific dilution factor was provided in the Mintel database, we used the same constant dilution factor for all powders to reflect the intrinsic nutritional quality of the product. However, an important drawback is that it does not reflect the nutritional quality of the product as consumed. To emphasize this limitation, we suggest to make the following changes :

Line 97: “standard” was changed to “constant”

Line 98: “since no product-specific dilution factor was provided in the food supply database” has been added.

Line 421-424: The following limitation was added to the discussion: “Moreover, when specific reconstitution guidelines are not provided for powdered products in the database, a constant dilution factor is applied to assess the intrinsic nutritional quality of the product. However, this might not fully reflect the actual nutrient content of powdered products when consumed.”

Line 87: what does the nutrient threshold means? May be daily intake %? Explain better… 

A precision on the unit of those nutrient threshold is indeed missing, we suggest addressing this point by making the following change:

Line 113: “The model generates nutrient thresholds in the reference unit, which are single values for each of the nutrients in the model”

Line 326: higher rather than higer  

Line 374: “higer” changed to “higher”

Figure 2A: What do you means “not available” (white bar) 

“Not available” designated data that were missing or not usable.

Figure 2A: “Not available” was changed to “missing” to add clarity

Appendix A: there is a technical mistake:  you have capital letters into the table and roman numbers in caption

Thank you for having spotted this discrepancy. We changed the title of the Appendix and the references to the appendix in the text (Lines 191, 225, 278, 388)

Reviewer 2 Report

This research is timely and needed for the food industry in the future.  However, there are number of issues that you'd need to address in the manuscript:

  1. The introduction is too brief. You should include a deeper description of nutrient profiling and include more recent references such as those in the Proceedings of the Nutrition Society August 2017 - e.g. M. Rayner, (pgs 230 - 236) and Adam Drewnowski, (pgs 220-229).  The Nutrition Society conference on Nutrient Profiling as a Tool to respond to Public Health Needs was held that year (2017) and you should include some of these references to demonstrate a thorough consideration of the subject.
  2. Line 62: Should read "With the help of new nutrient profiling models,  product reformulation could become a key strategy for health promotion at the population level.
  3. I suspect that 2. Materials and Methods was not written by the same person who wrote the intro and the discussion. That is fine, but there are the following problems with the english..:
    1. Line 78 - 79: You should be referencing this immediately. i.e. it should read: Criteria for upper and lower nutrient cut-points were selected based on Codex standards for 'low in' and 'source of' [6].
    2. Line79 - 80: "The values were selected from codex for saturated fats, total sodium and protein" - it's worth pointing out here that the EU labelling regulations now use salt rather than sodium.
    3. Line 89: Change to: "...,or must reach (lower limit) of a given nutrient value"
    4. Line 90: 'food supply database' - you must explain what you mean by this.
    5. Line 98: Change to: "Also excluded from model ratings" - doc currently reads 'form' 
    6. Lines 105 - 107: "Considering all entries......consumer offering" Not clear what is meant by this sentence.
    7. Line 107: Change to: "This categorization does not follow strict nutritional criteria"
    8. Line 112 - 113: "This type of categorization well serves the purpose of benchmarking.....consumer offering" Not clear what is meant by this sentence.
    9. Line 115: "cleaned database" - I understand that you mean that the data is clean, but the phrase 'cleaned database' is not routinely used.
    10. Line 116: Change to: "Cleaning consisted of removing products for which the sum of nutrient content values exceeded 100g."
    11. Line 124: Change to: "...for example Russia and Ukraine declare protein only..."
    12. Line 125: What do you mean by 'cold cereal' - must define terms like this.
    13. Line 126: Change to: "The highest number of protein omissions was also observed in the category 'carbonated soft drinks'."
    14. Line 127 - 129: "Since carbonated......any value)." Sentence does not make sense
    15. Line 130 - 132: This data should be put into a table.
    16. Line 143: "This is the case....." Are you saying 
      1. All nutrients are declared in g/100g or g/100ml OR
      2. are you saying that in some parts of the world some products are labelling /100g and in other parts of the world, the same products are labelled /100ml. Need to clarify.
    17. Line 189: Put in a sentence or two explaining why the heterogeneity is a problem. The detail in this para is well explained but a little clarity is required
    18. Line 205 - 206: Data should be tabulated.
    19. Line 236: "is visualised".....what do you mean by this?
    20. Line 268: Change to: "....and fruit and vegetables)."
    21. Line 281: "...missing in the database." While you have mentioned it before, it is worth referencing the Mintel Global New Products Database
    22. Line 325: spell individual correctly
    23. You have no Table 2 in the manuscript but you have Tables 1 & 3. You need to review this

Author Response

  1. The introduction is too brief. You should include a deeper description of nutrient profiling and include more recent references such as those in the Proceedings of the Nutrition Society August 2017 - e.g. M. Rayner, (pgs 230 - 236) and Adam Drewnowski, (pgs 220-229).  The Nutrition Society conference on Nutrient Profiling as a Tool to respond to Public Health Needs was held that year (2017) and you should include some of these references to demonstrate a thorough consideration of the subject.

Many thanks for your suggestion. We have expanded the introduction and added more than ten references to illustrate the variety of existing nutrient profiling systems and their applications.

  1. Line 62: Should read "With the help of new nutrient profiling models, product reformulation could become a key strategy for health promotion at the population level.

Line 82: “With the help of new nutrient profiling models, it can become a key strategy for health promotion at the population level” has been changed to “With the help of new nutrient profiling models, product reformulation could it can become a key strategy for health promotion at the population level.”

  1. I suspect that 2. Materials and Methods was not written by the same person who wrote the intro and the discussion. That is fine, but there are the following problems with the english..:
    1. Line 78 - 79: You should be referencing this immediately. i.e. it should read: Criteria for upper and lower nutrient cut-points were selected based on Codex standards for 'low in' and 'source of' [6].
    2. Line79 - 80: "The values were selected from codex for saturated fats, total sodium and protein" - it's worth pointing out here that the EU labelling regulations now use salt rather than sodium.

To be applicable globally, we aimed at selecting standards from codex when available. This explains the choice of sodium and not salt as a relevant nutrient for the study. It is now mentioned in the manuscript in the following way:

Line 100: “Whenever available in Codex, “low in” and “source of” limits were chosen for upper and lower nutrient cut-points [19]. This applies for saturated fats, total sodium and protein (with 50g as the nutrient reference value for protein). For sugars and fibers, European regulation was applied [20]”. The Codex Alimentarius is referenced sooner.

  1. Line 89: Change to: "...,or must reach (lower limit) of a given nutrient value"

Line 115-116: "...or must reach (lower limit) of a given nutrient value" was changed to "...or must reach a given nutrient value (lower limit). "

  1. Line 90: 'food supply database' - you must explain what you mean by this.

Line 116-118: “Thresholds are derived from a food supply database using a new algorithm that considers nutrients simultaneously, not independently “ was changed to “Thresholds are derived from a food supply database featuring nutrient composition of products; the new algorithm considers nutrients simultaneously, not independently.”

  1. Line 98: Change to: "Also excluded from model ratings" - doc currently reads 'form' 

Line 127: “form” was changed to “from”

  1. Lines 105 - 107: "Considering all entries......consumer offering" Not clear what is meant by this sentence.

Line 135-137: “Considering all entries from the previous three years ensures the use of representative up-to-date qualitative data that are grouped according to product categorizes representing the consumer offering” was changed to “Using entries from three consecutive years ensures up to date, qualitative and representative data that are categorized according to the consumer offering. “

  1. Line 107: Change to: "This categorization does not follow strict nutritional criteria"

Line 139: “This categorization is not necessarily following strict nutritional criteria.” Was changed to “This categorization does not follow strict nutritional criteria”

  1. Line 112 - 113: "This type of categorization well serves the purpose of benchmarking.....consumer offering" Not clear what is meant by this sentence.

Line 145: “This type of categorization well serves the purpose of benchmarking the nutritional quality of products competing within same consumer offering” Replaced by “This type of categorization enables the comparison of products within a same consumer offering”

  1. Line 115: "cleaned database" - I understand that you mean that the data is clean, but the phrase 'cleaned database' is not routinely used.
  2. Line 116: Change to: "Cleaning consisted of removing products for which the sum of nutrient content values exceeded 100g."

Line 149-152: “For the period 2016-2018, the cleaned database features 19 Mintel categories, 107 eligible Mintel food sub-categories, and 442,018 individual products. 100g Cleaning consisted in removing products for which the sum of nutrient content values per 100g was exceeding 100g” was changed to “Once cleaned, the database for the period 2016-2018 features 19 Mintel categories, 107 eligible Mintel food sub-categories, and 442,018 individual products. Cleaning consisted of removing products for which the sum of nutrient content values per 100g exceeded 100g”

  1. Line 124: Change to: "...for example Russia and Ukraine declare protein only..."

Line 161: “for example Russia and Ukraine declaring protein only “ Was changed to "...for example Russia and Ukraine declare protein only..."

  1. Line 125: What do you mean by 'cold cereal' - must define terms like this.

The quotation mark means that this is the category name given in Mintel Global New Product Database

  1. Line 126: Change to: "The highest number of protein omissions was also observed in the category 'carbonated soft drinks'."
  2. Line 127 - 129: "Since carbonated......any value)." Sentence does not make sense

Line 163-166: “This latter category was also the one with most missing protein content (18%). Since carbonated soft drinks are not expected to provide fibers or proteins, missing nutrient content were informative and considered as such in all product categories (i.e. neither set to zero, nor imputed to any value).” Was changed to “The highest number of protein omissions was also observed in the category 'carbonated soft drinks' (18%) which was expected since carbonated soft drinks usually do not contain fibers or proteins. Therefore, missing values are informative and considered as such in all product categories (i.e. neither set to zero, nor imputed to any value).”

  1. Line 130 - 132: This data should be put into a table.

Numbers are not important in those sentences; they are just mentioned as an example to illustrate the methodology. Therefore, we do not think they should be put forward in a table.

  1. Line 143: "This is the case....." Are you saying 
    1. All nutrients are declared in g/100g or g/100ml OR
    2. are you saying that in some parts of the world some products are labelling /100g and in other parts of the world, the same products are labelled /100ml. Need to clarify.

Line 184-189: For more clarity, “This is the case for ice-creams that are predominantly declared g/100g in America but g/100ml in Asia and Europe. It is also the case for reconstituted powders (e.g. dry soups, beverages) that are pre-dominantly declared g/100g powder in Asia and America and g/100ml as reconstituted in Europe.” Was changed to “This is the case for ice-creams for which all nutrients are predominantly declared as g/100g in America but g/100ml in Asia and Europe. It is also the case for reconstituted powders (e.g. dry soups, beverages) for which all nutrients are predominantly declared as g/100g of powder in Asia and America and as g/100ml of reconstituted beverage in Europe.”

  1. Line 189: Put in a sentence or two explaining why the heterogeneity is a problem. The detail in this para is well explained but a little clarity is required

Line 232-233: “Some of the 112 categories might still present some heterogeneity and might group multiple product offers.” Was replaced by “Some of the 112 categories are still nutritionally heterogeneous. This is problem because such categories group multiple consumer offering.”

  1. Line 205 - 206: Data should be tabulated.

Numbers are not important in those sentences; they are just mentioned as an example to illustrate the methodology. Therefore, we do not think they should be put forward in a table.

  1. Line 236: "is visualised".....what do you mean by this?

Line 245: “visualized” has been changed to “represented”

  1. Line 268: Change to: "....and fruit and vegetables)."

Line 315: "....and fruit and vegetable).” Was changed to "....and fruit and vegetables)."

  1. Line 281: "...missing in the database." While you have mentioned it before, it is worth referencing the Mintel Global New Products Database

Line 328: "...missing in the database." Was changed to "...missing in the Mintel Global New Products Database.”

  1. Line 325: spell individual correctly

Line 373: « indidividual » was replaced by “individual”

  1. You have no Table 2 in the manuscript but you have Tables 1 & 3. You need to review this

Line 351: Table 3 is now named Table 2.

Reviewer 3 Report

This paper outlines the main features of a method to monitor the nutritional quality of packaged foods to provide feasible targets for re-formulation and is a quite interesting for “Nutrients” readers.

Some modifications are suggested to the authors:

- Line 131: It is quite surprising that a food can contain 100g of protein per 100g of edible portion... You explain in the paper that some errors producing inconsistencies (e.g. summatory of main constituents should be equal or below 100) have been controlled, but it seems that you have not eliminated possible “outliers”. This is a serious drawback that you should discuss as a limitation of your study. By the way, include a paragraph at the end of the Discussion section where you discuss the main limitations of your study. I recommend that you to perform this data cleaning and repeat the analysis; this can be done in a short time.

- Results and Discussion section: You have not performed a validation study. You have compared your scoring method with one existing scoring method, but that doesn’t validate your method. So, any reference to “validation” should be eliminated. Additionally, you perform only the comparison with one scoring method, the Australian one, when you use data from around the world, so you should have compared your method with at least 3 scoring methods from different parts of the world, in order to check robustness. This is again a drawback that should be mentioned as a study limitation.

Author Response

- Line 131: It is quite surprising that a food can contain 100g of protein per 100g of edible portion... You explain in the paper that some errors producing inconsistencies (e.g. summatory of main constituents should be equal or below 100) have been controlled, but it seems that you have not eliminated possible “outliers”. This is a serious drawback that you should discuss as a limitation of your study. By the way, include a paragraph at the end of the Discussion section where you discuss the main limitations of your study. I recommend that you to perform this data cleaning and repeat the analysis; this can be done in a short time.

Thank you for raising this interesting point. In the Mintel database, 2% of products have more than 30g/100g and among those, only 12 products have 100% proteins: one gelatin product and meal replacements targeting sportsmen. Both can be reformulated; for instance, some manufacturers add usually add carbohydrates to gelatins. As this nutrient composition does not come from incorrect data but reflects the intrinsic nutritional quality of two specific categories, this small number of products should not be defined as outliers and thus, were included in the analysis.

Results and Discussion section: You have not performed a validation study. You have compared your scoring method with one existing scoring method, but that doesn’t validate your method. So, any reference to “validation” should be eliminated. Additionally, you perform only the comparison with one scoring method, the Australian one, when you use data from around the world, so you should have compared your method with at least 3 scoring methods from different parts of the world, in order to check robustness. This is again a drawback that should be mentioned as a study limitation.

Thank you for this interesting comment. As there is no “gold standard” to validate nutrient profiling systems, the choice was made to provide a relative validation to an accepted methodology: Health Star Rating, like it has been done in previous nutrition studies.

We suggest adding the following sentence to the discussion to stress this point: “This high correlation of the new method with HSR provides first evidence that it is valid relative to an accepted methodology [29]”. – Lines 443-445

Reviewer 4 Report

The manuscript deals with the development of a new method to monitor the nutritional quality in terms of the nutrient composition of pre-packaged foods. The developed algorithm is based on the 5 key nutrients (i.e., sugar, saturated fats, sodium, fibre and proteins). The data were retrieved from the Mintel’s Global New Products Database and the aim of the developed nutrient profile system was to be used for product reformulation. The manuscript is well written, and the basis for developing the algorithm well described with examples that will help readers to understand what they did. Finally, the new score is validated against the Health Start Rating with a strong agreement between the two methods. This reviewer applauds the quality of the manuscript.

Author Response

We are very pleased to see that you appreciated the methodology developed and do not request any further changes. Thank you very much.